# Challenges of Preadolescence in the School Context: A Systematic Review of Protective/Risk Factors and Intervention Programmes

**Maria Lidia Mascia ***[ID]**, Giulia Langiu, Natale Salvatore Bonfiglio, Maria Pietronilla Penna**[ID] **and Stefania Cataudella**

Department of Pedagogy, Psychology, Philosophy, University of Cagliari, 09123 Cagliari, CA, Italy
\* Correspondence: marialidia.mascia@unica.it

**Abstract:** Preadolescence is a critical period, characterised by changes in physical, hormonal, cognitive, behavioural, and emotional development, as well as by changes in social and school relationships. These changes are accompanied by the transition from elementary school to middle school. The literature shows that this transition is one of the most stressful events for preadolescents, which can have a negative impact on their well-being. The main objectives of this review, focused on the school context, were to identify protective and risk factors influencing the well-being of preadolescent students and to describe the interventions implemented. A systematic search of peer-reviewed papers published between 2011 and 2021 was conducted following the PRISMA reporting guidelines. A total of 36 full-text articles met the inclusion criteria. Studies converge in identifying risk factors that may affect student well-being in this age group: individual factors (levels of emotional awareness and self-esteem) and relational factors (friendship, teachers' and parents' supporting actions and roles). Intervention programs are mainly focused on improving emotional and social regulation skills that also influence academic achievement. Our findings have important implications for both research and intervention in school settings.

**Keywords:** preadolescents; middle schools; risk factors; protective factors; intervention programs

## 1. Introduction

Preadolescence is characterised by simultaneous changes in physical, hormonal, cognitive, behavioural, and emotional development, as well as by changes in social and school relationships [1]. These changes feature in the first transition preceding the adolescent period which involves challenges distinct from those of childhood and adolescence. Preadolescence, however, is not just a time of transition but deserves specific attention as a phase of life with all its specific characteristics. Preadolescence is proposed as a distinct period where changes that we observe more manifest in adolescence begin to take shape and begin to appear: a sense of internal decompensation and increased conflict in relationships, specifically, with peers and with parents [2]. The physical and hormonal changes are the first big changes in preadolescence, and they coincide with sexual maturation [2,3]. Physical development is evident in both sexes: girls are prone to breast development, hair grows in the pubic area and armpits, menstruation, height growth, and hormone changes can increase oil made by the skin that may cause acne; in boys, there are evident voice changes, the rapid growth of the trunk, hair growth in the pubic area, on the arms and legs and in armpits, and penis growth and the first involuntary erections [4]. Biological changes have important psychological and social impacts that influence preadolescents to accept themselves physically [5]. Changes that occur during puberty give rise to new emotional experiences such as a sense of insecurity or feeling nervous [6]. Additionally, the fluctuation of hormonal levels and brain development can affect mood and create mood swings and cause difficulties in emotional regulation: as a result, preadolescents show behavioural

patterns such as parent–child conflicts, sensation seeking, and the development of romantic interests [7]. Preadolescence is, therefore, considered a period of vulnerability that increases the risk of the emergence of problems, above all in the relational and emotional field [8,9].

From the psychological point of view, this life season is characterised by the identification process, which means that preadolescents begin to identify with their peers and start to conflict with reference figures (parents and teachers)—comparison and identification with their reference group is the most important goal in this development period. The building of their individuality and identity starts from this stage of life [10]. Preadolescents live in an important conflict between the safety of the world of childhood and the drive to begin to create their own identity. Preadolescents start to redefine their relationship with their parents and begin to attach more importance to symmetric relationships than asymmetric ones [11,12].

These changes in social relationships are accompanied by the transition from elementary school to middle school, almost everywhere in the world. Indeed, although school systems have different nomenclatures and classifications, it is agreed that there is a transition precisely in this age group. The different school systems of different nations refer to the same band and years of schooling even for different systems (see Figure 1).

| AGE | USA Canada Australia India | Italy | UK | Brazil | Japan | |
|---|---|---|---|---|---|---|
| 3 | EC1 | asilo | Nursery | Infantil | | Early childood |
| 4 | EC2 | asilo | Reception | Jardim | | |
| 5 | Kindergarten | asilo | Year 1 | Jardim | | |
| 6 | Grade 1 | 1a elem | Year 2 | Ensino f | Grade 1 | Elementary |
| 7 | Grade 2 | 2a elem | Year 3 | Ensino f | Grade 2 | |
| 8 | Grade 3 | 3a elem | Year 4 | Ensino f | Grade 3 | |
| 9 | Grade 4 | 4a elem | Year 5 | Ensino f | Grade 4 | |
| 10 | Grade 5 | 5a elem | Year 6 | Ensino f | Grade 5 | |
| 11 | Grade 6 | 1a media | Year 7 | Ensino f | Grade 6 | Middle School |
| 12 | Grade 7 | 2a media | Year 8 | Ensino f | Grade 7 | |
| 13 | Grade 8 | 3a media | Year 9 | Ensino f | Grade 8 | |
| 14 | Grade 9 | 1 liceo | Year 10 | Ensino f | Grade 9 | High School |
| 15 | Grade 10 | 2 liceo | Year 11 | Ensino f | Grade 10 | |
| 16 | Grade 11 | 3 liceo | Year 12 | Ensino i | Grade 11 | |
| 17 | Grade 12 | 4 liceo | Year 13 | Ensino i | Grade 12 | |
| | | 5 liceo | | Ensino i | | |

**Figure 1.** Different school systems classification.

This period of school plays a crucial role in educational development, transitions in the relational system both with peers and teachers, in the autonomy required, in the approach to study, etc.; students experience both a contextual change (schools) and a personal transition (puberty).

The extant literature shows that transition in middle school is one of the most stressful events for preadolescents that negatively impacts their self-concept, self-esteem, and academic achievement [13–15]. Eccles et al. [16] identified the typical middle school or junior high environment as developmentally inappropriate for preadolescents. The drastic switch in the school environment is difficult for most students to manage. As students attempt to negotiate the contextual change, an intense personal change often occurs with the beginning of puberty. These normative changes create important challenges for students during transition. In the new school scenario, sometimes the student's history is not known, and it is difficult to get in touch with students; sometimes this is because the system does not provide transition support. There is also a transition in the class group, which must re-form and establish a new relationship and peer support system [14,17]. Roeser et al. [18] report that preadolescents' perceptions of academic competence, the valuing of school, and emotional health are all important predictors of students' grades, conduct in school, and the quality of their peer relationships. A recent review [19] examined findings concerning the impact of the primary to secondary education transition on both psychological and academic outcomes, and they noted that individual difference factors such as cognitive and emotional ability levels, gender, and socio-emotional skills can moderate the association between young adolescents' academic motivation and engagement, self-concept, affect toward school, and their intrinsic interest in school; additionally, the review highlighted the importance of the social support received from parents, teachers, and peers in helping students to feel more secure and socially accepted during the transition experience.

Although preadolescence represents a very delicate period and one to which continuous attention must be paid, one of the problems is to identify the boundaries between preadolescence and adolescence development periods, and this has led to an underestimation of this phase, which has fluid and sometimes undefined boundaries. It is important to think about this age group because we can glimpse some precursors that may later develop into problem behaviours. Many papers about preadolescence even present a gap of about two years in indicating the beginning and end of preadolescence. As well as overlap and confusion about the target age group, this confusion about boundaries can also be seen in the nomenclature used, with some articles referring to childhood [20], others to early adolescence [21], pre-teens [22], to young adolescence [23], and others directly to adolescence [24]. The issues and characteristics related to preadolescence are not new in the literature; formerly, Blair and Burton [25] described the period of later childhood, dwelling on physical and intellectual transformations, on the influence of social context in this development, above all to give suggestions for parents in guidance. The variables that are intertwined in preadolescence and the complexity that characterises it led researchers to constantly question who preadolescents are, how they experience change, and what, if any, precursors need to be monitored and acted upon with preventive or supportive interventions.

For this reason, it is important to constantly analyse the most recent literature in this area so that we can grasp the speed of change in an ever-changing world, providing a framework that will aid researchers in evaluating new literature and give new directions for future research.

In this paper, we have conducted a systematic review concerning preadolescents with special reference to the middle school context.

The main objectives of this systematic review were: to identify individual and relational factors influencing the well-being of preadolescents in the school context; to identify specific protective and risk factors related to the school transition period; and to describe the interventions implemented in the middle school context.

## 2. Materials and Methods

### 2.1. Overview

This study was conducted following the Preferred Reporting Items for Systematic Review and Meta-Analyses (PRISMA) guidelines [26].

### 2.2. Research

PsycInfo, PsycArticles, Scopus, PubMed, and the Psychology & Behavioral Science Collection were searched systematically, using the following keywords: "preadolescence" AND ("psychology mental health" OR "wellbeing" OR "psychology risk factors*" OR "psychology protective factors"). We focused on peer-reviewed articles published from 2011 to 2021 inclusive. Results were limited to English, Italian, and German language peer-reviewed journal publications. Primary searches were completed in December 2021.

### 2.3. Inclusion and Exclusion Criteria

To be included in this review, studies must have participants aged 9 to 14 recruited in the school setting; the administration of the protocols and any intervention programmes must have been delivered within the school setting.

Outcomes must be measured through quantitative and/or qualitative methods. Book chapters, dissertations, meta-analyses, reviews, comments, letters, editorials, and theoretical papers were excluded.

Studies were also excluded if they were: case studies; instrument validation; medicalfocus; socialfocus; biological/organic/genetic/neurological factors; articles with adult perspective on preadolescents; articles on sleep regulation; physiological factors and stress; and articles with borderline age range (over 9–14).

As regards school setting, studies were included if they: (i) considered emotional and social development; (ii) considered risk and protective factors related to mental health in preadolescents; (iii) evaluated the effects of intervention programmes that promote well-being in school; and (iv) treated transition problems.

### 2.4. Study Selection and Extraction Steps

All identified citations were imported into the bibliographic manager software Zotero 5.0 (Corporation for Digital Scholarship, Virginia, USA). Duplicates were identified and removed, after which, abstracts and titles were screened by three independent reviewers (SC, GL, and MLM) for eligibility. Discordant eligibility determinations were resolved by consensus.

The full texts of the eligible records were then obtained and screened for eligibility according to the exclusion criteria. Any doubts or conflicts were resolved by discussion between the three reviewers (SC, GL, and MLM) to reach a consensus.

### 2.5. Study Selection and Extraction Steps

Three independent reviewers (SC, GL, and MLM) created a data extraction standardised form in Microsoft Word (Microsoft Corporation, Redmond, WA, USA). The first author, year of publication, number, gender and age of participants, study setting, variables, protocols, test, intervention programmes, and outcomes were extracted from each of the studies included. Any discrepancies in the extracted data were resolved by discussion between the four reviewers (SC, GL, NSB, and MLM) to reach a consensus.

## 3. Results

### 3.1. Study Selections and Extractions

A total of 4240 abstracts were located of which 1527 were removed, being duplicates or not articles. A total of 2713 studies were screened against titles and abstracts. Subsequently, 2468 studies were excluded, principally because they did not fit the inclusion criteria and 47 studies were about school settings. A total of 36 studies finally met our inclusion criteria and, concerning the school context, were included (see Figure 2).

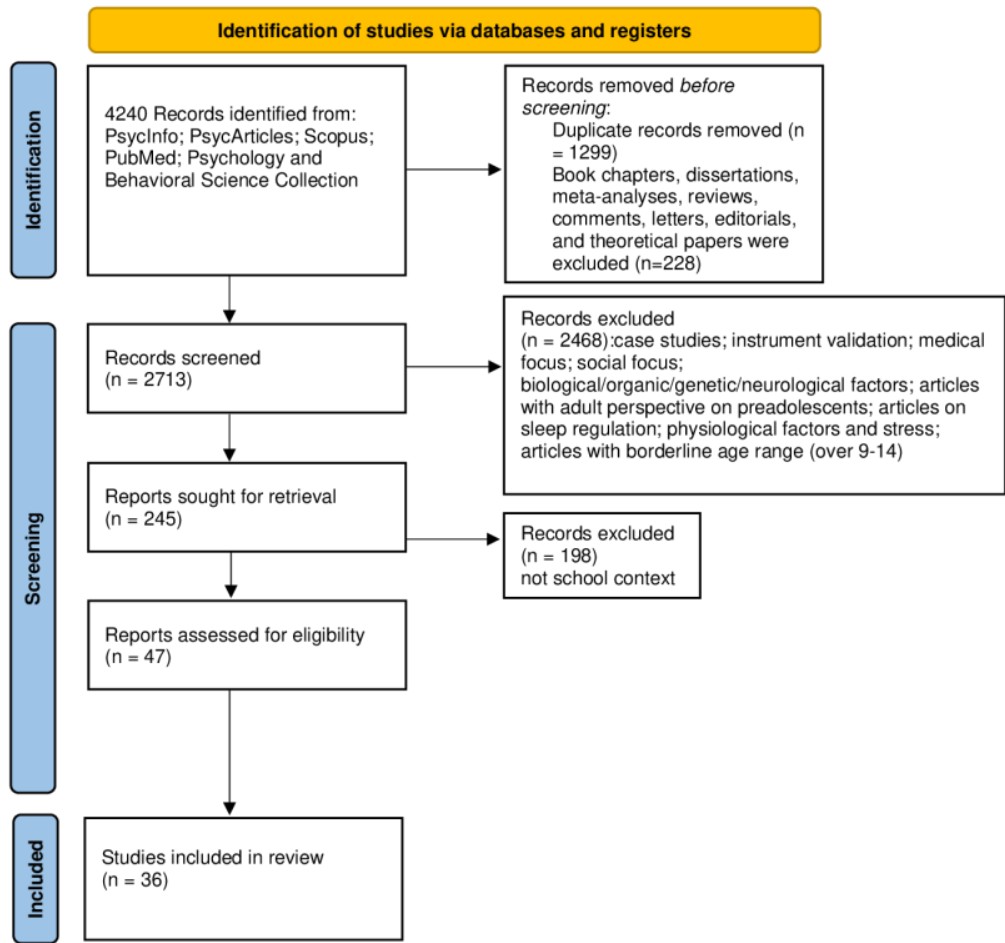

**Figure 2.** PRISMA Scheme.

*3.2. Studies Characteristics*

We summarized the key results of the study characteristics in Table 1 and S1 in the Supplementary Material.

The total of subjects in all selected studies was 35,669. Participants were all students attending the last year of primary school or middle school, while only one study of the longitudinal type considered the first year of secondary school.

Participants were principally students observed in the school context. The students considered had specific characteristics; in some cases, they present a disability (n = 3 studies).

Only a few studies (n = 9) used specific intervention programmes.

As regards methodology and study design, some of the selected studies used follow-up or a pre- and post-evaluation after intervention assessment (n = 15).

Most of the studies divided samples into subgroups (e.g., gender, separate conditions) (n = 24). Of these, ten studies have control groups or conditions and nine have reported results for control groups/conditions.

Only a few studies have selected conditions (n = 6) or subjects (n = 7) randomly. Thirteen used only validated measures, and five used only non-validated measures. No studies used a placebo condition.

**Table 1.** Characteristics of publications included in review (N = 36).

| N. | Study (36) | Participants: N, Mean Age (SD), Country | Issue Investigated | Variables and Measures | Projects | Summary of Main Results |
|---|---|---|---|---|---|---|
| 1 | (Amado-Alonso et al., 2018) [27] | 840 participants; 494 boys and 346 girls. Age: 9–12. Spain | Relationship between the hours of organized sports practice and self-concept from a multidimensional approach (physical, emotional, academic, social, and family self-concept) | Hours of organized sports practice and AF-5 Self-Concept Form 5 questionnaire. Gender function. | No | The findings suggest that organized sports practice could have a positive effect on self-concept |
| 2 | (Arens et al., 2013) [28] | 625 participants; (45.3% boys and 64.7% girls; M age = 10.21; SD = 0.59) Age: not indicated. German | Effects of transition to secondary school in Germany on students' self-perceptions | Self-Description Questionnaire I (SDQ I; Marsh, 1990). | No | The results showed that 5th graders had lower levels of self-esteem compared to 4th graders. 4th graders were found to have higher mean scores in both the competence and affect components of all facets of academic self-concept examined here |
| 3 | (Ashrafi et al., 2020) [29] | 3829 participants; boys and girls (not indicated) Age: 9–14. Canada (Saskatchewa) | Bullying victimization among preadolescents | Student Health Survey. Variables: bullying victimizations, socio–demographics, behavior, psychological, and relationship variables. | No | Students in younger grades were bullied more often than students from older grades; greater victimization occurring in lower grades, where peer relations and peer group structure play a more salient role; suicide ideation in the past year was significant among studentsaggressively victimized; Self-perception of being overweight is yet another factor that is more common in the moderately and aggressively victimized subgroups; drinking behavior was also found to be significantly associated with peer victimization; preadolescents who had a stronger social network with peers and better relationships with their parents were less likely to be bullied because they could confide in their peers or parents when they faced challenges |

**Table 1.** *Cont.*

| N. | Study (36) | Participants: N, Mean Age (SD), Country | Issue Investigated | Variables and Measures | Projects | Summary of Main Results |
|---|---|---|---|---|---|---|
| 4 | (Attar–Schwartz et al., 2019) [30] | 669 participants; boys and girls (not indicated) Age: 9–14. Canada | The Role of Classmates' Social Support, Peer Victimization and Gender in Externalizing and Internalizing Behaviors among Canadian Youth | Self-report questionnaires: internalizing and externalizing behaviors; bullying victimization experiences; social support; socio-demographic information. | No | Our results show relatively high levels of reported perceived support by the three sources that were examined. They show, however, the highest perceived support received from parents and the lowest from classmates. This finding may show that despite the possible decreased dependence on parents they remain significant figures in adolescents' lives, and while classmate support is less significant than that of parents, it is still high and important |
| 5 | (Baerveldt et al., 2014) [31] | 741 participants; (50.2% girls and 48.8% boys) Age: 12–14. Holland | Selection patterns, gender, and friendship aim in classroom networks | Information about friendships was collected by means of a nomination procedure; Activity level; Linking, Deep similarity. | No | Rapidly changing friendship patterns in the beginning of a school year can have an interference with academic achievement and personal well-being |
| 6 | (Caprara et al., 2014) [32] | 151 participants; (52.3% males; M age 12.4), and the control group 173 students (50.3% females; M age 13.0). Italy | Beneficial effects of prosocial behavior in adolescence | Variables: Agreeableness (AGR); Empathic Self-Efficacy Beliefs (ESE); Prosocial behavior; academic achievement. | Yes: school-based intervention called CEPIDEA | A significant increase of helping behavior along with a decrease in physical and verbal aggression across time; higher grades than the control group at the end of middle school |
| 7 | (Chen et al., 2019) [33] | 913 participants; averaged 9.9 (SD = 0.30) and 11.9 (SD = 0.34) years of age at the Grade 4 and Grade 6 time points, respectively. United States | Understand individual characteristics that may predict friendship quality during this developmental period | Attribution Biases and Friendship Quality | No | A hostile attribution bias concurrently predicted lower self-reported friendship quality only for girls; they may be more likely to withdraw from positive interactions when they perceive a friend's actions as hostile |

**Table 1.** *Cont.*

| N. | Study (36) | Participants: N, Mean Age (SD), Country | Issue Investigated | Variables and Measures | Projects | Summary of Main Results |
|---|---|---|---|---|---|---|
| 8 | (Clemmensen et al., 2020) [20] | 1170 participants; aged 11–12 years. Denmark (Copenaghen) | Theory of mind (ToM) and bullying separately associated with later academic performance among preadolescents? | Intelligence; Academic achievement; Bullying | No | Effect of bullying and ToM on academic performance among adolescents and found bullying |
| 9 | (Coelho et al., 2017) [15] | 1147 participants; aged 9 to about 12 years (M age = 9.62; SD = 0.30), from six public–school groupings. Portugal | Promoting a Positive Middle School Transition: A Randomized Controlled Treatment Study Examining Self-Concept and Self-Esteem | Self-Concept; Self-Esteem | Yes: Positive transition program– project designed to support student transition from 4th grade (the final grade of elementary school) to 5th (the first grade of middle school in Portugal) | The results of the current study reveal that students report lower levels of academic, emotional, and physical self-concept, as well as lower levels of self-esteem by the end of 5th grade in relation to 4th grade students. No gender differences. Participation in the Positive Transition program seems to have lessened the negative impact of middle school |
| 10 | (Coelho and Sousa, 2017) [34] | 982 participants; (Mage = 11.22; SD = 1.02, 46.8% girls): 318 in the control condition, 355 students. Portugal | Efficacy of Middle School Social and Emotional Learning Program | Social and Emotional Competencies: Bateria de Socializac¸a~o 3 (BAS–3; Portuguese adaptation by Ferreira and Rocha 2004): five dimensions. Self-esteem questionnaire | Yes: Positive Attitude— designed to improve children's social and emotional competencies by helping them develop social awareness, self-awareness, self-control, relationship skills, and responsible decision making | The program led to gains in several social and emotional competencies: namely, social awareness, self-control, and self-esteem. It contribuite to reduce social isolation and social anxiety. |

Table 1. *Cont.*

| N. | Study (36) | Participants: N, Mean Age (SD), Country | Issue Investigated | Variables and Measures | Projects | Summary of Main Results |
|---|---|---|---|---|---|---|
| 11 | (Coyle et al., 2021) [35] | 799 participants; (51.8% male), 7th and 8th grade students from one suburban middle school in the Midwest | Find patterns of bullying involvement | The multi–construct victim survey (MCVS; Demaray & Malecki, 2003) andan additional item to measure cyberbullying in addition to the items assessing verbal, physical, and relational bullying, and victimization. Behavior Assessment System for Children, Second Edition (BASC–2; Reynolds & Kamphaus, 2004) Self Report of Personality (SPR). | No | Results from the current study support prior research that has found bully–victims to be the group at greatest risk for internalizing problems. The bully–involved groups demonstrated significantly more internalizing, externalizing, and school-related problems than youth not involved in bullying |
| 12 | (de La Haye et al., 2017) [36] | 714 participants; (N = 504 trials participants) and the non–participating classmates that they nominated (N = 210) | How overweight was associated with relations of friendship and dislike (antipathies) in the peer group | Friendships, antipathies, Anthropometry, demographic | No | Social environment, characterized by fewer friendships and greater antipathies contribute to put overweight youth at increased risk for psychosocial maladjustment |
| 13 | (Delgado et al., 2019) [37] | 548 participants; (M age = 10.95, SD = 0.7), with 275 boys (50.2%) and 273 girls (49.8% girls). Spain | The relationship between self-concept, academic goals, and the participation of the roles of victim, bully, and bystander in cyberbullying by gender and grade | Self-concept (Self-Description Questionnaire I), academic goals (Achievement Goals Tendencies Questionnaire), cyberbullying (Cyberbullying; Screening for peer bullying and cyberbullying), gender, and grade | No | Five predictive models for the role of victim are found |

**Table 1.** *Cont.*

| N. | Study (36) | Participants: N, Mean Age (SD), Country | Issue Investigated | Variables and Measures | Projects | Summary of Main Results |
|---|---|---|---|---|---|---|
| 14 | (Dorio et al., 2019) [38] | 887 participants; (53.7% male) sixth-through eighth-grade students from a middle school in northern Illinois | Association between peer victimization and school engagement and the indirect effects of rumination and depressive symptoms in this association | Traditional victimization; Cyber victimization; School engagement; Depressive symptoms; Rumination | No | Poor school engagement may result from victimization experiences activating or triggering students' levels of rumination and symptoms of depression |
| 15 | (Dursley et al., 2015) [39] | 276 participants (116 female and 161 male); participants were recruited from a primary school (133 children, M age = 9.92 years, SD age = 0.72) | Perceived seriousness of disruptive classroom behaviours | Perceived seriousness of disruptive classroom behaviours; social behaviour; social desirability | No | Children who regarded disruptive classroom behaviour are more serious engaged in more positive behaviour |
| 16 | (Dvorsky et al., 2018) [23] | 93 participants; (67 males, 26 females) with ADHD initially assessed when in fifth (39.9%), sixth (31.1%), or seventh (29.0%) grade (age range 10.4 to 14.3) | Protective Effects of Social Factors on the Academic Functioning of Adolescents with ADHD | Parent and adolescent report of social skills and social acceptance and multiple academic outcomes were evaluated longitudinally, including grades and teacher ratings of academic impairment. ADHD and ODD symptoms. Social Skills; Social Acceptance; School Grades; Teacher-rated Academic Impairment | No | Social acceptance moderated the association between inattention and grades; high parent- and parent- and adolescent- rated social acceptance served as a promotive factor but not as a protective factor for teacher-rated academic impairment |

**Table 1.** *Cont.*

| N. | Study (36) | Participants: N, Mean Age (SD), Country | Issue Investigated | Variables and Measures | Projects | Summary of Main Results |
|---|---|---|---|---|---|---|
| 17 | (Fairweather–Schmidt and Wade, 2015) [40] | 125 participants; (47.2% girls) (mean age 11.60 years, SD = 0.82; range 9.91–13.91) in upper primary school classes from three independent schools in Adelaide, South Australia | Evaluate a school-based intervention program focusing on reducing perfectionism in preadolescent children. | The Child and Adolescent Perfectionism Scale; The Strength and Difficulties Questionnaire; Questionnaire for the development of disordered eating | Yes: school-based intervention program | Preliminary support for the effectiveness of an intervention focused on reducing levels of perfectionism in children |
| 18 | (Hoffmann et al., 2020) [21] | 1540 participants; (9–14 years of age, 733 females). Brazil | Independent and interactive associations of temperament dimensions with educational outcomes | Early Adolescence Temperament Questionnaire; Educational outcomes: counting suspension, repetition, and dropout events, parent reports on overall academic performance and by reading and writing standardized tests | No | High effortful control and fear were independently associated with better educational measurements; high levels of frustration and surgency were independently associated with worse educational measurement. Effortful control and frustration interact, such that low frustration and low effortful control were a detrimental combination for poor reading ability |
| 19 | (Ickovics et al., 2013) [41] | 940 participants; (grades 5 and 6). New Haven, Connecticut | Underlying behavioral risk factors for chronic disease—nutrition, physical activity, and smoking—assessed within the social and environmental context in which people live, work, and attend school | Health index was constructed to include 14 diverse, modifiable, and important health assets from 4 domains: physical health, health behaviors, family environment, and psychological well-being | No | Creative approaches that integrate curricular and non-curricular school-wide efforts to promote healthy behaviors among all students are worth the investment |

**Table 1.** *Cont.*

| N. | Study (36) | Participants: N, Mean Age (SD), Country | Issue Investigated | Variables and Measures | Projects | Summary of Main Results |
|---|---|---|---|---|---|---|
| 20 | (Layous et al., 2012) [42] | 415 participants; (M age = 10.6). Vancouver, Canada. | Importance of Prompting Prosocial Behavior in Increasing Preadolescents Boosts Peer Acceptance and Well-Being | Satisfaction (Satisfaction with Life Scale adapted for children), happiness (Subjective Happiness Scale adapted for children), and positive affect (child version of the Positive and Negative Affect Schedule) | Yes: perform three acts of kindness (versus visit three places) per week over the course of 4 weeks | Doing good for others benefits the givers, earning them not only improved well-being but also popularity |
| 21 | (Long et al., 2012) [43] | 921 participants; (M age = 12.70; SD = 0.68) included 7th (50.1%) and 8th (49.5%) grade students | The tripartite model of subjective well-being (SWB) | Multidimensional Students' Life Satisfaction Scale; the Positive and Negative Affect Scale for Children | No | A 4-factor model comprised of positive emotions, negative emotions, fear-related negative emotions, and SS best described the structure of school-related SWB in the current sample |
| 22 | (Makover et al., 2019) [44] | 2664 participants; six middle schools completed a universal emotional health screening during the second half of the 8th grade year. Pacific Northwest | This study examined the impact of a school-based indicated prevention program on depression and anxiety symptoms for youth during the transition from middle to high school | Depressive Symptoms; Aggression and demographic characteristics | Yes: High School Transition Program—prevention program on depression and anxiety symptoms | The HSTP intervention evidenced a small to moderate effect for depression symptoms and a small effect size for anxiety symptoms |
| 23 | (Murray and Zvoch, 2011) [45] | 193 participants; African American youth from low-income backgrounds U.S.A. | Relationship between teachers and students | Child Report Measures and Teacher Report Measures | No | Both student and teacher perceptions of teacher–student relationship quality were associated with student- and teacher-rated emotional, behavioral, and school-related adjustment |

| N. | Study (36) | Participants: N, Mean Age (SD), Country | Issue Investigated | Variables and Measures | Projects | Summary of Main Results |
|---|---|---|---|---|---|---|
| 24 | (Nitkowski et al., 2017) [46] | 543 participants; (46.6% girls, mean age = 11.46 years, SD = 0.69). German | Exploring the relation between subjective well-being, emotional awareness, and emotion expression | Subjective Well-Being, Emotional Awareness, and Emotion Expression | Yes:Emotion Training with Students (ETS) | The negative impact of emotional awareness on subjective well-being was diminished |
| 25 | (Ogurlu et al., 2018) [47] | 117 participants; 44 female (37.6%) and 73 (62.4%) male; 51 (43.6%) students were 5th grade, 30 (25.6%) students were 6th grade, 31 (26.5%) students were 7th grade, and 5 (4.3%) students were 8th grade students. Turkey | Relationship between social–emotional learning skills and perceived social support of gifted students | Social Emotional Learning Skills Scale (SELSS) and Child–Adolescent Social Support Scale (CASS) | Yes: afterschool enrichment program | Participants scored the highest on the frequency section from the close friends and the lowest points were from the classmates. Theyconsidered social support from teachers as the most important in this study. Gender differences in perceived social support were found. |
| 26 | (Ojio et al., 2019) [22] | 662 participants; grade 5 to 6 students from nine elementary schools. Japan | Effects of a school teacher-led 45-min educational program for mental health literacy in pre-teens | Questionnaire about knowledge about mental health/illnesses, recognition of mental health state of a character in a vignette; recognition of the necessity to seek help; intention to seek help (self); intention to help (peers) | Yes: teacher–led program for mental health literacy (MHL), suitable for schools with tight schedules | School teacher-led program had positive effects on MHL in pre-teens |

**Table 1.** *Cont.*

| N. | Study (36) | Participants: N, Mean Age (SD), Country | Issue Investigated | Variables and Measures | Projects | Summary of Main Results |
|---|---|---|---|---|---|---|
| 27 | (Olivier and Archambault, 2017) [48] | 513 participants; fourth- to sixth-grade students (50.5% girls) in seven elementary schools. Canada | Whether closeness with teachers and prosociality toward peers protect students displaying hyperactive or inattentive behaviors against behavioral, emotional, and cognitive disengagement throughout the school year | Student engagement; Hyperactivity and inattention; Prosociality toward peers; Close student–teacher relationship; Gender and age; Academic achievement; Parental school support | No | Prosociality is a protective factor against inattentive students' risk of disengagement. Prosocial students' tendency to team up and to benefit from each other's positive influence, which fosters their behavioral engagement in school. |
| 28 | (Rothon et al., 2011) [49] | 4887 participants; 48.6% of the participants were male. London, England | The extent to which social support can have a buffering effect against the potentially adverse consequences of bullying on school achievement and mental health | Bullying; Educational achievement; Depressive symptoms; Social support; Ethnicity; Free school meals | No | There was evidence that a high level of support from friends was able to protect bullied adolescents from poor achievement at school. A moderate (but not high) level of support from the family was also protective |
| 29 | (Rusby et al., 2013) [50] | 82 participants of seventh grade (36 at risk for developing or escalating rule-breaking and substance use, and 46 randomly selected) from four schools participated Oregon, U.S.A. | Describe the associations among perceptions of peer affiliates, mood states, and social contexts | Ecological Momentary Assessment (EMA) to simultaneously capture youths' perceptions of peer affiliates and social contexts to determine their association with youths' current and future mood states | No | Happiness was associated with affiliating with peers who were perceived as popular. Negative moods were associated with affiliating with peers by whom they are teased or treated meanly |

Table 1. *Cont.*

| N. | Study (36) | Participants: N, Mean Age (SD), Country | Issue Investigated | Variables and Measures | Projects | Summary of Main Results |
|---|---|---|---|---|---|---|
| 30 | (Salazar–Ayala et al., 2021) [51] | 1132 participants in the fifth and sixth grades in public elementary schools, ages between 10 and 13 (M = 10.51 years; SD = 0.66 years). Chihuahua, Mexico | Correlation and predictive relationship between controlling teaching and the fear of negative evaluation mediated by the frustration of the basic psychological needs (BPN), controlled motivation, and individualism/competitiveness | Controlling Teaching Style; BPN frustration Psychological Needs Frustration Scale in Physical Exercise; Controlled Motivation; Individualism/Competitiveness; Fear of Negative Evaluation | No | There is a predictive relationship between a controlling teaching style over the fear of negative evaluation with the mediators FBPN, controlled motivation, individualism, and competitiveness |
| 31 | (Schaffhuser et al., 2017) [24] | 248 participants; (average age at T1 = 10.6 years). Basle, Switzerland | The development of global and domain specific self-representations in the transition from late childhood to early adolescence and tested whether gender, puberty, and school transition help explain individual differences in change | Self-representations; Pubertal status; Pubertal timing; School transition. | No | Significant mean–level decreases of global self-esteem and the academic and physical self-concepts from late childhood to early adolescence. It seems that the participants generally saw themselves in a more negative light in terms of self-esteem when they moved to early adolescence |
| 32 | (Schuster et al., 2012) [52] | 5119 participants; randomly selected public school fifth graders and their parents in three towns: Birmingham, Houston, and Los Angeles | Racial and Ethnic Health Disparities | 16 measures, including witnessing of violence, peer victimization, perpetration of aggression, seat–belt use, bike–helmet use, substance use, discrimination, terrorism worries, vigorous exercise, obesity, and self-rated health status and psychological and physical quality of life. | No | Significant differences between black children and white children for all 16 measures and between Latino children and white children for 12 of 16 measures, although adjusted analyses reduced many of these disparities. |

**Table 1.** *Cont.*

| N. | Study (36) | Participants: N, Mean Age (SD), Country | Issue Investigated | Variables and Measures | Projects | Summary of Main Results |
|---|---|---|---|---|---|---|
| 33 | (Tomada et al., 2015) [53] | 628 participants; fifth-graders (Male = 319, Female = 309; M age = 10 years and 11 months). Italy | The emotional closeness of the teacher and dissatisfaction with his or her behavior influence scholastic skills and performance of the students. Relationship between relational variables and the different ways of relationship with peers | Emotional closeness and dissatisfaction; scholastic skills and performance; prosocial and aggressive behavior; acceptance and rejection. | No | The influence of closeness has little relevance on academic success; in males, unlike in females, the "normative" dimension has an influence, albeit a modest one, on academic success, while the affective dimension is completely irrelevant in the process in question |
| 34 | (Troop-Gordon et al., 2019) [54] | 484 participants; (239 girls; M age = 10.25 years). U.S.A. | Child characteristics and relationship qualities that predict pro-bullying bystander behavior over the course of one school year | Pro-bullying bystander behavior and bullying behavior; Empathy; Moral disengagement; Peer victimization; Popularity; Perceived norms for defending. | No | Significant mean-level decreases of global self-esteem and the academic and physical self-concepts from late childhood to early adolescence. It seems that the participants generally saw themselves in a more negative light in terms of self-esteem when they moved to early adolescence |
| 35 | (Vaz et al., 2014) [55] | 266 participants; (197 typically developing students and 69 students with a disability). Australia | Explore and compare perceived AC and MHF of students with and without disability, six months before and six months after transition to secondary school | Self-Perception Profile for Adolescents (SPPA); MHF: The Strengths and Difficulties Questionnaire (SDQ); Family demographics and school contextual characteristics. | No | A significant reduction in the contribution of personal background factors on AC subsequent to the transition, despite AC scores staying stable across time; students with a disability had lower AC than their typically developing peers. An improvement in the disability subgroup's AC is underlined after the transition |

**Table 1.** *Cont.*

| N. | Study (36) | Participants: N, Mean Age (SD), Country | Issue Investigated | Variables and Measures | Projects | Summary of Main Results |
|---|---|---|---|---|---|---|
| 36 | (Vaz et al., 2015) [56] | 266 participants; (53.4% girls) with and without disabilities who negotiated the transition from 52 primary schools to 152 secondary schools (M age at T1 = 11.89, SD = 0.45; and at T2 = 12.9, SD = 0.57). Australia | Determine whether students' perceptions of school belongingness changes across the primary–secondary school transition; determine which factors associated with belongingness in primary school, continue to be associated with belongingness in secondary school, and if they maintain their influence; determine whether there are additional factors and, if so, to develop the best–fit model of belongingness in secondary school | Instruments to measure: coping skills, perceived competence, expectation, motivation, and orientation schooling, mental health functioning. Contextual factors: family factors, school and classroom factors, school belongingness. | No | The findings highlight the need for primary and secondary schools to organise classroom goals, tasks, and assignments, and foster pluralism among all students to promote school belongingness |

## 4. Discussion

This systematic review focused on preadolescents' age starting from the school context, looking to understand what kind of factors can be risky or protective in preadolescents attending middle school. What is immediately clear is the multitude of variables examined by the studies read and analysed. Although many studies used validated measures, only a few used rigorous methodological conditions such as control groups or conditions for comparison. Moreover, no studies used more than two control conditions, and only a few studies randomised subjects or conditions. For these reasons, results must be interpreted with caution.

It is important to reflect on some specific areas and focuses that require careful consideration, either to understand how to act preventively or to intervene. The findings on developing positive behaviours represent a common point among the papers examined. Promoting preadolescents' well-being represents a complex challenge for parents, teachers, and the education world in general. Acting on preadolescence and individual and contextual variables is a key point to be able to help young people in their first outlook on life in an early form of autonomy and peer comparison. Preadolescents' well-being is associated with many aspects of the relationship with parents, family, peers, friends, education, school, and academic performance.

Some of the main themes are presented in this review; the topics covered often intertwine and overlap, and the same study may present several themes at once. However, we found four principal kinds of factors: factors related to personal, cognitive, and individual aspects; factors related to relational aspects; factors related to the transition period to middle school; and factors related to support made by projects and programmes and specific figures such as teachers.

### 4.1. Aspects Related to Individual Factors

Individual factors such as personal and cognitive aspects can have a great influence on preadolescents' lives and general well-being.

Long et al. [43] explore subjective well-being, given by frequent positive emotions, infrequent negative emotions, and a positive evaluation of life circumstances [57] looking to extend the model of social well-being to the specific context of youth and their schooling. They found a 4-factor model comprising positive emotions, negative emotions, fear-related negative emotions, and school satisfaction. An important function is represented by emotional awareness. The relationship between subjective well-being, emotional awareness, and emotional expression is confirmed by other studies [46]. Emotional awareness is also included in many programmes for students [15].

Among individual factors, as indicated by Coelho et al. [15] and Coelho and Souza [34], a crucial role is represented by self-esteem levels as predictors of well-being among students. Self-esteem is a factor to be monitored and enhanced, especially in the transition from childhood to adolescence. Preadolescents often face a decrease in their self-esteem, especially in the transition period.

Amado-Alonso et al. [27] concentrate on self-concept and self-representation. Self-concept is one of the most important outcomes of the processes of education and socialisation, and it is described as the perceptions that people have about themselves, formed through the interpretation of their own experience and of the environment, particularly influenced by significant others' reinforcements and feedback, as well as by their own cognitive mechanisms [27]. Self-concept is a multidimensional construct which brings together academic, social, family, emotional, and physical dimensions [34]. In their study, Amado-Alonso et al. [27] underline this last dimension stating that organised sports practice could have a positive effect on self-concept in the preadolescent period.

Troop-Gordon et al. [54] state that significant mean-level decreases in academic and physical self-concepts occur from late childhood to preadolescence.

*4.2. Aspects Related to Relational Factors*

Children's socialisation (above all between ages 6 and 12) is promoted because children not only develop in the family environment, but they also begin to get to know their peers, which allows them to relate to them through games, interactions, and so forth. Their participation in new situations and the link between affectivity and the environment will contribute to forming the child's personality [34]. Many studies in this review focus on the advantages of socialisation for children and preadolescents. A body of literature identifies friendship as a protective factor in many areas of preadolescent life. In this period of life, a friendship develops and changes its balance, and its role becomes crucial for all the effects related to it, e.g., it can interfere with academic achievement and personal well-being [31], in particular, the authors show the results of the rapid changes in friendship patterns at the beginning of a school year.

Chen et al. [33] look to understand individual characteristics that may predict friendship quality during this developmental period, and they find that a hostile evaluation bias leads to a lower quality of friendship but only for girls. De la Haye et al. [36] concluded that overweight preadolescents have an elevated risk of psychosocial maladjustment when surrounded only by a small number of friendships.

Positive relational aspects have a fundamental role as a protective factor for preadolescents' risk behaviours. This element is underlined in most of the papers' focus on risks connected to bullying, which is extremely prevalent in this age group [35]. Victimisation experience, the other face of the bullying phenomenon, can lead to poor school engagement and high rumination levels and depressive symptoms [38]. Students in younger grades were bullied more often versus students from older grades; greater victimisation occurred in the lower grades, where peer relations and peer group structure play a more salient role [29]. Positive relational aspects among peers are fundamental to prevent all risks allied to victimisation in bullying behaviours [58].

Delgado et al. [37] found an interesting model that connects self-concept, academic goals, and the participation of the roles of victim, bully, and bystander in cyberbullying, showing how low scores in social self-concept and academic self-concept can explain cyber victimisation and the incidence of cyberbullying. Moreover, the authors reported that social self-concept with peers and learning goals can be protective factors against acting as a perpetrator.

A protective factor in this kind of risk is given also by teachers' and parents' supporting actions and roles [37,53]. Murray, Harvey, and Slee [45] found that a high level of stressful relationships and a low level of supportive relationships with family, teachers, and peers were associated with greater bullying, victimisation, and psychological health problems, as well as poorer social/emotional adjustment. In the preadolescent period, it is also sensible to provide education on the importance of relationships in life. Caprara et al. [32] underline the beneficial effects on preadolescents' life of prosocial behaviours. The study by Layous et al. [42] shows that doing good to others leads to greater well-being and popularity. In their study, they demonstrate increased happiness and acceptance through prosocial activity. This has benefits for both the preadolescent and the community; well-liked preadolescents exhibit more inclusive and less externalising behaviour (e.g., less bullying) as adolescents. Other authors underline the importance of acting to promote inclusion and prevent racism or racial and ethnic disparities [45]. A fundamental role in the relationship's foundations is played by classmates' social support [46].

*4.3. Factors Related to the Transition Period to Middle School*

Some of the articles identified [15,24,44,55] dealt with the transition to "middle school", while others focus on the transition to high school [44,55].

The results and reflections presented in these papers focus on some important concerns that students can manifest during school transition periods because, typically, the school transition involves simultaneous changes in life, the school environment, relationships, and academic expectations. The problems associated with transition are united by the fact

that the student needs support and help to cope with a period full of changes. Coelho et al. [15] show that preadolescent students report lower levels of academic, emotional, and physical self-concept as well as lower levels of self-esteem by the end of fifth grade (1st year of U.S. middle school) compared to the fourth grade (last year of elementary school). As Coelho et al. [15], other research on this topic takes into account that this period of transition can present a decline in academic self and self-esteem attributed to individual and environmental factors; only Arens et al. [28] attribute this decline primarily to social and contextual factors. No gender differences are considered significant in all studies on transition considered in this review.

Makover et al. [44] focus on depression and anxiety symptoms and show, once again, the effectiveness of intervention programmes, in this case, aimed at students facing the transition from middle school to high school and, more specifically, from preadolescence to adolescence. The targeted intervention programme seems to reduce the escalation of depression, anxiety, and associated school difficulties in a group of U.S.A. at-risk youth transitioning from middle to high school. Results indicated that in comparison to the control condition, the group who followed the programme showed a small moderate effect for depression symptoms and a small effect size for anxiety symptoms.

This review takes into account two studies from Vaz [55,56] on the effects of the transition on the experiences of typically developing students or those with a disability, trying to understand the strengths and difficulties encountered by the two groups examined but also the influence of background. Individual student factors and primary school contextual factors are more important contributors in post-transition adjustment than concurrent secondary school contextual factors, and there exists a greater responsibility on primary schools to ensure that the transition needs of the disadvantaged groups are met satisfactorily.

Research from Vaz [55,56] allows us to reflect on how important it is to structure transition pathways that ensure continuity and consider all types of students.

*4.4. Factors Related to Support Made by Projects and Programmes and Specific Figures Such as Teachers*

This age group appears receptive and if accompanied by the right support preadolescents welcome the cues to be able to improve and grow. It is against this backdrop that programmes organised primarily in the school setting, with different goals of helping the pre-teen, assume great importance. For example, the purpose of many programmes is to improve social skills, which, in this age group, are precisely those that begin to be built with the skills of self-regulation, both emotional and social, which also influence academic achievement. An interesting and original study by Fairweather-Schmidt et al. [40] presents a programme to give preliminary support for the effectiveness of an intervention focused on reducing levels of perfectionism. This makes us reflect on the fact that there are so many factors related to well-being on which we can intervene early.

These interventions play a key role because they provide the preadolescent with attention, make them feel welcome, and are very often structured to ensure motivation for action, present and future. Such programmes, therefore, also aim to ensure support.

For example, through participation in a programme called the "Positive Transition", Coelho et al. [15] promote school adjustment in the transition to middle school referring to the achievement of higher levels of self-concept representation and self-esteem.

The quality of social support plays a crucial role in the development of preadolescents, including their self-concept, mental health, academic achievement, social development, and the risk of victimisation and bullying. In the school context and this age group, in addition to their parents and family context, key support is generated in the relationship with the teacher. Olivier and Archambault [48] state that the closeness with teachers and prosociality towards peers protect students displaying hyperactive or inattentive behaviours against behavioural, emotional, and cognitive disengagement throughout the school year.

## 5. Conclusions

From our review, we identified 36 articles published over a 10-year period. This leads us to reflect that perhaps more attention should be paid to the psychological aspects that characterise this stage of life as a specific age. Instead, we observed that this age is predominantly equated with childhood studies or adolescent studies. However, this age group deserves a great deal of attention, above all, because it allows for preventive actions to be taken both to identify risk factors and to address protective factors or growth factors to foster effective development or the strengthening of all those skills that can contribute to the individual's well-being, both personal and social. In particular, from the analysis of the articles taken in account in this review, we have identified four factors which the research has prioritised: personal, cognitive, and individual; relational; transition period problems; and support made by projects and programmes and by specific figures such as teachers. In this age group, the self is forming, and the individual has yet to become autonomous and must learn how to relate to themselves and others. The purpose of much research is to identify factors on which preventive action can be taken to be able to curb or avoid problems related to the individual's present and future well-being. To this end, the aim of research is to understand how to help the construction of the self, how to make sure that the preadolescent has a positive self-representation, and what can be the supporting figures and elements that can accompany them on this path of discovery and growth. Literature presents also many of the programmes created, for example, to improve social skills that precisely in this age group are those that begin to be built, and the skills of self-regulation, both emotional and social, also influence learning. The focus of other programmes is to contrast bullying and cyberbullying, phenomena which are increasing globally, specifically in recent years, and have dangerous impacts on students' lives and are allied to some negative aspects, such as moral disengagement [59]. It is fundamental to support preadolescents above all in the school context. Some research shows that positive, warm, and supportive relationships between the teacher and student are critical for healthy and positive development, including in terms of academic achievement, social skills, and in countering bullying [60,61]. For example, sometimes student–teacher conflict, defined as a type of student–teacher interaction characterised by a perception of mutual discontent, disapproval, and unpredictability [62], can impact student life beyond schooling.

A topic considered is also school transition, always worthy of attention, in all age groups. Developmental changes and challenges in preadolescence provide a unique opportunity for society; problems related to transition, above all in this moment of preadolescents' lives, highlight the need for society and the schools to organise classroom goals, tasks, and assignments, and promote collaboration and help among all students in such a way as to give school belongingness and continuity [44].

It is essential to understand who preadolescents are in order to identify the factors that promote well-being, especially in the school context where preadolescents spend most of their time. It is fundamental to provide valuable insights into whether educators and other professionals in the field should focus on addressing developmental issues related to puberty or on environmental and class climate issues related to the teaching and learning processes at secondary school during this critical stage of life transition.

**Supplementary Materials:** The following supporting information can be downloaded at: https://www.mdpi.com/article/10.3390/educsci13020130/s1. Table S1: Classification of publications included in review (N = 36).

**Author Contributions:** Conceptualization, M.L.M., G.L. and S.C.; methodology, M.L.M., G.L., N.S.B., M.P.P. and S.C.; investigation, G.L.; data curation, M.L.M., G.L., N.S.B. and S.C.; writing—original draft preparation, M.L.M., G.L., N.S.B., M.P.P. and S.C.; writing—review and editing, M.L.M. and S.C.; supervision, S.C. All authors have read and agreed to the published version of the manuscript.

**Funding:** This review received no external funding.

**Institutional Review Board Statement:** Not applicable.

**Informed Consent Statement:** Not applicable.

**Data Availability Statement:** Not applicable.

**Conflicts of Interest:** The authors declare no conflict of interest.

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
