# Peer review of "Challenges of Preadolescence in the School Context: A Systematic Review of Protective/Risk Factors and Intervention Programmes"

_education, doi:10.3390/educsci13020130_

Round 1

Reviewer 1 Report

The abstract section should be more specific, it is too generic. Keywords should be checked. The introduction section is well explained and well founded, but figure 1 is not provided as indicated in the third paragraph of the text, so it cannot be assessed. In the penultimate paragraph of this section the objective of the work is included and it is not very clear, for possible readers, if the objective is the systematic review, or to detect the challenges of the pre-adolescent in the school context, or to analyze the risk factors that they can affect the development of adolescents or, as the text says, help the family, educational and school systems to intervene in improving the well-being of adolescents. This aspect should be better specified.

In the methodological section, the general description is very brief (it does not provide any scheme to verify the steps followed with the PRISMA model); also synthesis point 2.6 is too brief. In the results, figure 2 is not provided, which prevents the assessment. The other aspects of the analysis of results and discussion are well presented and substantiated and fit the title and objective of the work.

The conclusions section contains comments on the subject, but not on the procedure and the systematic review (only one final sentence). This is an aspect to improve.

Author Response

Dear Reviewer,

Thank you for your letter Ref. Manuscript ID: education-2139780-R1, entitled “Challenges of preadolescence in the school context: a systematic review of protective/risk factors and intervention programmes”, and for giving us the opportunity to review and resubmit the paper.

We are very grateful to your and the reviewers’ comments and suggestions; we are deeply appreciative of your careful reading.

Detailed replies to your comments are enumerated below, with the list of modifications and integrations. We hope this revised version now satisfies the requirements for publication in your journal.

Then, we submit the revised version of paper; for clarity new portions, added or modified in response to the referees’ comments, are highlighted in the manuscript; furthermore, the tracked version of the manuscript is attached.

Thank you very much

The Authors

Reviewer 2 Report

I sincerely appreciate the opportunity to review this paper. I have read and reviewed the manuscript with great interest.

This manuscript is a systematic review focusing on protective/risk factors and the effects of interventions in schools. The scope of this study seems appropriate; Adolescence is a life stage at the intersection of human bio-psycho-social maturation, and mental health during preadolescence and adolescence has received increasing attention in recent years. The importance of providing appropriate mental health education, especially in schools, the center of young people's social life and an essential place for interaction with peers, has been repeatedly pointed out.

This study employed a reliable and standard methodology based on PRISMA; the methods and results of this study are described in an appropriate manner that follows the PRISMA checklist. In addition, based on the reviewed contents, this study pointed out the limitations of previous studies (e.g., the need for more structured and randomized studies), also providing recommendations for future research.

This systematic review is considered worthy of publication; however, there are some major and minor concerns with this study, pointed out below.

Major concerns.

#M1: The 1st paragraph of the Introduction

In the manuscript, the definitions of each life stage (childhood, preadolescence, and adolescence) need to be clarified. In particular, the definition of "preadolescence" should be explicitly mentioned so that the difference from childhood/ adolescence is clearly understood.

In addition, the scope of this study, including its title, is declared as "preadolescence," but not a few papers reviewed in the manuscript include "adolescence." Previous studies on adolescence sometimes include teenagers and children after puberty (roughly 10-12 years of age).

Please clarify the definition of "preadolescence" and revisit whether it is consistent with the previous studies reviewed.

#M2: The last paragraph of the Conclusions

The manuscript stated that future studies should examine the impact of the COVID-19 pandemic or the role of excessive mobile phone use. These are not implications that can be derived directly from the content itself of this study. The reasons for focusing on COVID-19 and mobile phones need to be described persuasively based on the review results of this study.

Minor concerns.

#m1: The 1st paragraph of the Introduction

Citations No.2 and No.3 are cited from the AAP's website. These are secondary references (as stated at the bottom of the pages). It will be more appropriate to mention the primary literature.

#m2: The 2nd paragraph of the Introduction

In the part where Erikson's developmental tasks are mentioned, "the fourth stage ("industry")" is contrasted with "the fifth stage ("fidelity")." This may be a confusion between the Psychosocial Crisis in each stage and the Basic Virtue to be acquired, described by Erikson (Stage 4, Psychosocial Crisis = "Industry vs. Inferiority," Basic Virtue = Competency; Stage 5, "Identity vs. Role confusion," Fidelity). It would be better to re-state the concepts to provide an appropriate contrasting structure.

#m3: 2.1. Overview

The version of the PRISMA guidelines (2020) should be clearly stated. It should also be added to the citation.

I hope that the above comments will be of some benefit in improving the manuscript.

With appreciation,

Author Response

(The authors gave the same response as above.)

Reviewer 3 Report

Thank you for your thorough review, which focuses close on the pre-adolescent life stage. Figure 2 is referred to on line 170 but does appear in the paper. The inclusion of this table seems unnecessary. I think that you make your case for educational research to recognise pre-adolescence as a distinct life stage.

Author Response

Dear Reviewer,

Thank you for your letter Ref. Manuscript ID: education-2139780-R1, entitled “Challenges of preadolescence in the school context: a systematic review of protective/risk factors and intervention programmes”, and for giving us the opportunity to review and resubmit the paper.

We are very grateful to your and the reviewers’ comments and suggestions; we are deeply appreciative of your careful reading.

Thank you very much

The Authors

Round 2

Reviewer 2 Report

Journal: Education Science

Article Title: Challenges of preadolescence in the school context: a systematic review of protective/risk factors and intervention programmes

I sincerely appreciate the authors' sincere responses to my review comments. The resubmitted manuscript was quite adequately revised. I think the manuscript is of sufficient quality for publication. However, I would like to suggest one minor revision.

Minor concerns.

#m1: The 1st paragraph of the Introduction

The revised citation #3 is a public-relation declaration that recommends avoiding clinically unrecommended practices in medical practice. It does not seem an appropriate citation that directly explains associations between "hormonal surges" and "brain maturation or psychological/behavioral development" at the beginning of puberty.

I recommend that citation #3 be replaced with another appropriate one. I think many appropriate articles can be found by searching for "adolescence, puberty, hormone, review." It would be appropriate to cite a section for pubertal hormonal changes from pediatric or endocrinology textbooks since hormonal changes during puberty are fully established knowledge.

I hope that the above comments will be of some benefit in improving the manuscript.

With appreciation,

Author Response

Thank you for your appreciation and for your precious suggestions. We have amended and replaced the quotation, accepting your recommendations.

Thank you again for your time.
